# Durability and Self-Sealing Examination of Concretes Modified with Crystalline Waterproofing Admixtures

**DOI:** 10.3390/ma14216508

**Published:** 2021-10-29

**Authors:** Pejman Azarsa, Rishi Gupta, Peiman Azarsa, Alireza Biparva

**Affiliations:** 1Department of Civil Engineering, Facility for Innovative Materials and Infrastructure Monitoring (FIMIM), University of Victoria, 3800 Finnerty Rd., Victoria, BC V8W 3R4, Canada; pazarsa@uvic.ca (P.A.); azarsap@uvic.ca (P.A.); 2Kryton International Inc., 1645 East Kent Ave N, Vancouver, BC V5P 2S8, Canada; alireza@kryton.com

**Keywords:** crystalline waterproofing admixtures (CWA), durability of cement-based materials, permeability reducing admixtures (PRA), freeze–thaw resistance, steel reinforcement corrosion, self-sealing

## Abstract

Repairing concrete structures costs billions of dollars every year all around the globe. For overcoming durability concerns and creating enduring economical structures, chemical admixtures, as a unique solution, have recently attracted a lot of interest. As permeability of a concrete structure is considered to play a significant role in its durability, Permeability Reducing Admixtures (PRA) is one of the ideal solutions for protecting structures exposed to water and waterborne chemicals. Different products have been developed to protect concrete structures against water penetration, which, based on their chemistry, performance, and functionality, have been categorized into PRA. As it has previously been tested by authors and proven to be a promising solution, a hydrophilic Crystalline Waterproofing Admixtures (CWA) has been considered for this study. This paper aims to investigate how this product affects concrete’s overall freeze–thaw resistance, self-sealing, and corrosion resistance. Various testing methods have been utilized to examine the performance of CWA mixtures, including the linear polarization resistance, resonance frequency testing, half-cell potential, and self-sealing test. The reinforcement corrosion potential and rate measurements indicated superior performance for CWA-treated samples. After being exposed to 300 freeze–thaw cycles, concrete mixes containing CWA—even non-air-entrained ones—showed a Durability Factor (DF) of more than 80% with no signs of failure, while non-air-entrained control samples indicated the lowest DF (below 60%) but the greatest mass loss. The major causes are a reduction in solution permeability and lack of water availability in the concrete matrix—due to the presence of CWA crystals. Furthermore, evidence from the self-sealing test suggests that CWA-treated specimens can seal wider cracks and at a faster rate.

## 1. Introduction

Deterioration in reinforced concrete structures can occur because of various stresses and deformations from internal, mechanical, or environmental loadings. This deterioration allows water, aggressive ions, or gases to easily penetrate concrete, intensifying its transport properties and resulting in reduced durability. Premature durability issues (corrosion of reinforcing steel, freezing and thawing cycles, etc.) shorten the service life of concrete structures and increase the cost of rehabilitation, which include indirect costs related to the socioeconomic and environmental implications of these actions [1]. In this setting, the durability of concrete structures has become a top priority for owners, stakeholders, and governments.

### 1.1. Crystalline Waterproofing Admixtures (CWA): Background

Owing to its porous nature, one of the key parameters connected to the durability of concrete structures is its resistance to water permeability. Despite the fact that a properly proportioned and cured concrete with a low *w*/*c* ratio can ideally yield a durable finished product, it is commonly accepted that no concrete material or structure can be made waterproof or “bottle tight” [2]. As a result, concrete professionals must consider the following factors to control porosity and reduce permeability in order to create a “watertight” concrete structure: low *w*/*c* ratio and the use of chemical/mineral admixtures; proper cement content; aggregate gradation; quality manufacturing processes; and careful execution of joints and penetrations [3]. 

Various types of chemical admixtures, known as Permeability-Reducing Admixtures (PRAs), have become developed in recent decades to particularly meet these requirements for improving concrete durability by increasing the watertightness of concrete. PRAs can be classified based on the service conditions of the concrete structure being exposed, according to American Concrete Institute (ACI) TC 212 [4]: PRAH, or Permeability-Reducing Admixture for Hydrostatic conditions, and PRAN, or Permeability-Reducing Admixture for Non-hydrostatic conditions. The capillary absorption transport mechanism should be considered in a structure with non-hydrostatic pressure; however, for concrete structures under hydrostatic conditions, water permeability is the primary transport mechanism to consider [5]. PRA materials typically include: hydrophobic water repellents, polymer products, finely divided solids, hydrophobic pore blockers, and crystalline products [4]. Hydrophobic or water repellents admixtures, as well as inert and/or chemically active fillers, are proposed as PRANs suitable for damp proofing in non-hydrostatic conditions [5]. Crystalline Waterproofing Admixtures (CWA), on the other hand, is a PRA sub-category that can be applied as both PRAN and PRAH, according to the ACI TC 212 report [4]. They are often employed to protect structures against moisture, aggressive ions, and subsurface water. Although different PRAs are designed to lower permeability, their chemistry and chemical interactions varies, which might cause them to behave differently in concrete from one admixture to another. Hence, the performance of one type of PRA admixture stated in the literature does not always guarantee that all other PRAs will perform equally. 

### 1.2. Effects of CWA on Concrete Properties

The performance and benefits of CWA treatment had been investigated and demonstrated in some previous research studies [6,7,8,9,10,11,12,13,14,15,16,17,18]. Žižková et al. [8] stated that the curing conditions during the first 28 days of ageing are an important factor that influences the CWA’s effectiveness. In a research study by Žižková et al., they also came to the conclusion that some CWA had an impact on the amount of Portlandite (CH) produced [10]. After exposure to aggressive environments, SEM sampling revealed differences in the microstructure of mortars with and without CWA [10]. They also claimed that the CWA reduced the apparent porosity of the mortar greater than control mixes. The results of Pazderka and Hájková’s laboratory testing of the speed of the CWA waterproofing effect revealed that a CWA modified concrete structure might be theoretically ready for applying water pressure on the 12th day after creation [6]. In the presence of CWA materials, different phases of Calcium Silicate Hydrate (C–S–H) were produced, according to Elsalamawy et al.’s study [12]. With a Ca/Si ratio ranging from 2.4 to 3.2, these phases range from spherical particles to needle-like crystals. 

### 1.3. Effects of CWA on Concrete Self-Sealing

In addition to lowering concrete permeability, when it comes to designing a durable, water-tight concrete structure, almost all durability design codes, standards, and guidelines assume that the concrete will remain un-cracked throughout its service life [19]; however, cracking in concrete is unavoidable due to many possible reasons, such as plastic shrinkage and excessive tensile stresses. The two primary causes of reinforcement corrosion and subsequent deterioration of concrete structures are aggressive ions ingress and carbonation, which are both accelerated by cracking [20]. In order to avoid structures from degrading in real life, cracks must be repaired early as possible [20]. However, due to their location and/or environmental conditions, it is not always cost-effective or convenient for concrete professionals to access damaged sites for repair work. Hence, the concrete community is becoming more interested in self-sealing (SS) crack technology. As one of the desired solutions, the effectiveness of crystalline waterproofing materials (CWA) in terms of crack sealing has also gained substantial attention from researchers and engineers in this industry. Under water immersion conditions, Jaroenratanapirom and Sahamitmongkol studied the visual closure of cracks in mortar specimens incorporating fly ash, expansive admixtures, silica fume, CWA, and limestone powder [21,22]. Sisomphon et al. investigated the self-sealing potential and water tightness of pre-cracked cement-based materials treated with Calcium Sulfo-Aluminate (CSA) based expansion additive and CWA [23,24]. Ferrara et al. [25] studied the CWA effects on the concrete’s SS and their healing capability on the recovery of mechanical properties; evaluated the influences of the SS phenomena on the recovery of stiffness and load-bearing capacity by means of 3-point bending test before and after conditioning [25]. Roig-Flores et al. [26] investigated the effects of CWA solely on the SS of concrete in four types of environmental exposure conditions using a different technique. They developed a method for evaluating the SS properties of cracked samples based on the specimen’s global permeability and distinct geometrical characteristics of the crack before and after the SS period. Following up on the previous study, Roig-Flores et al.’s [27] work looked into the SS properties of early-age concrete that had been engineered with CWA by measuring the permeability of cracked samples and their crack width. Ferrara et al. [28] also investigated the effect of CWA on the self-sealing capacity of the cementitious composites using both a Normal Strength Concrete (NSC) and a High-Performance Fiber-Reinforced Cementitious Composite (HPFRCC). The presence of sealing products on the sealed surfaces was confirmed by SEM observations and EDS analyses in Cuenca et al.’s work, owing to delayed hydration and carbonation reactions involving both the cement and the CWA [29]. Furthermore, the crack sealing efficiency of CWA mixes combined with Super Absorbent Polymer (SAP) was quite satisfactory in Li et al.’s [30] study.

### 1.4. Effects of CWA on Concrete Properties: Authors Previous Findings 

The performance of a hydrophilic CWA is investigated in this study. This admixture is made up of active chemicals in cement and sand carriers, which are mixed with the other ingredients in the concrete and typically used when the structure is subjected to hydrostatic pressure (PRAH) [4]. The waterproofing effect of CWA concrete is primarily due to the formation of needle-shaped crystals within its pore structure [4]. Performance of concretes containing stated hydrophilic CWA is investigated for enhancing durability and resisting hydrostatic water pressure by researchers including the authors. According to the Robertson study et al., the concrete panels containing similar hydrophilic CWA as the one in this work, performed well, showing low half-cell values and no apparent evidence of corrosion after 10 years of exposure in a real-field corrosive environment [31]. Authors also performed various testing and evaluated different durability properties of CWA concrete [13,14,15,18,32,33]. The authors’ findings of assessing the permeability and self-sealing of CWA concrete with two different common cement types revealed that adding CWA to the concrete mix reduced the water permeability coefficient by three times while increasing the self-sealing ratio at a faster rate [13,14,18,33]. The cementitious composites with CWA utilized in this investigation showed needle-like crystal formation, which is different from ettringite, based on another authors study results [15]. Other two CWA admixtures in the same study—claimed by other manufacturers to have the same qualities as the one in the study—revealed sulfur peaks in Energy Dispersive Spectrum (EDS) graph, similar to ettringite [15]. The average porosity of mixtures containing CWA was also found to be lower than ordinary mix in SEM images and X-ray examinations [15]. This emphasizes the fact that different PRA and CWA classes have diverse chemistry and chemical formulations, implying that their performances will differ. The test results also indicated that the CWA can effectively reduce the early-age shrinkage cracking [33]. 

### 1.5. Methodological Approach and Research Significance

Overall, the value of CWA as a PRA in the construction industry has already been recognized; however, after reviewing the literature on CWA concretes, some inconsistencies in results were noted, prompting further research in this area. Additionally, since different PRA and CWA have distinct chemistry, we might expect variable performance in virgin concrete mixtures; hence it has yet to be determined whether CWA have effects on some durability characteristics. To the best of authors’ knowledge, no studies have discussed the concrete freeze–thaw resistance and reinforcement corrosion rate for modified mixtures containing CWA—hence, this is the first. Using a well-established and appropriate laboratory test method, limited experimental studies have been performed to determine the SS capability of CWA concrete. As a result, the objective of this work is to explore the efficacy of CWA in enhancing SS mechanisms, improving the reinforcement corrosion potential, and freezing/thawing resistance characteristics of concrete in an experimental setting. Reinforcement corrosion properties were determined using a Half-Cell Potential (HCP) test as described in ASTM C876 [34], a macro corrosion test in accordance with ASTM G109 [35], and Linear Polarization Technique (LPR). Concrete freezing and thawing cycles resistance and dynamic modulus of elasticity have also been determined using ASTM C666 [36] and ASTM C215 [37], respectively. The obtained experimental test results are both qualitatively and quantitatively analyzed, compared with past related studies that used similar materials, and eventually drew some scientific conclusions. The SS properties of cracked specimens are also evaluated using a well-standardized methodology based on both microscopic/visual examination and measurement of the flow rate through cracks, as described in the authors’ prior work [13,33]. The significance of producing more durable and sustainable concrete structures that can benefit all construction sectors and stakeholders by saving on costs and time for the lifecycle of a project, is indeed high. Hence, this study is intended to demonstrate that the use of CWA can significantly contribute to developing durable concrete elements. However, because they are all different in chemistry and chemical reactions, designers should choose the most appropriate admixtures based on their particular needs.

## 2. Experimental Methodology

The materials, mixture proportions, casting procedure, curing conditions, and specimen preparation are all included in this section, as well as the testing procedures used to determine the self-sealing and specific durability properties of untreated and CWA-treated concrete.

### 2.1. Materials and Mixture Proportions

Portland cement-type 10 (also known as Type GU in CSA A23.1-19 [38]) with a 0.53 *w*/*c* ratio was utilized for concrete mixtures in accordance with ASTM C150 [39] (Table 1). CWA, in powder form, was added to the concrete mixtures at a dosage of 2% by weight of cement. Generally, CWA consists of a proprietary mix of active chemicals implanted in a carrier of cement and sand that react with cement compounds (like admixture used in this study) or by-products of cement hydration, most likely Ca(OH)_2_ (other types) as described by the American Concrete Institute (ACI) Committee 212.

In the study by Sisomphon et al. [23], it was stated that calcium hydroxide (CH) is the reactive component when tested for a type of CWA. As a result of the chemical reaction and deposition of integrally bonded crystals into the hardened cement paste, pressure resistance of modified matrix increases as high as 14 bars [4]. The chemical compositions of these products are proprietary and not available; however, some of the chemical and physical properties of the material, reported by its manufacturer and used in this study, are given in authors past studies [13,15]. In compliance with ASTM C150 [39], Quikrete Portland cement-type 10 (also referred as Type GU in CSA A23.1-14 [40]) was used for concrete mixtures in this study. Naturally available fine aggregate, meeting ASTM C33 [41] requirement, from Sechelt pit in B.C., Canada was used for the experimental work. Moisture content was measured by the change in weight of sample after keeping the sample for 24 h in an oven at 100 °C. The calculated moisture content was 1.35%. The fineness modulus, density and absorption of fine aggregate were 2.60, 2.65 and 0.79%, respectively, in accordance with ASTM C127 [42].

### 2.2. Specimen Preparation

All of the mixtures were cast according to ASTM C192 [43] specifications, and they were tested for compressive strength [44], slump [45], and air content [46] in compliance with ASTM standards. Three concrete beams per mixture were prepared using standard procedure described in ASTM G109 [35] to conduct the reinforcement corrosion tests. Figure 1 shows that the specimen is 280 × 150 × 115 mm (11 × 6 × 4.5 in.) in size, with two rebar 25 mm (1 in.) from the bottom and one rebar at the top, with the distance between its top and the top surface of the specimen being twice the maximum aggregate size (20 mm in this case). The middle of the rebar was exposed about 200 mm (8 in.) and the rest was covered with electroplater’s tape. A 90 mm (3.5 in.) rebar length was then protected with neoprene tubes over the electroplater’s tape at each end of the bar (Figure 2). Specimens were removed from the forms and cured for 28 days before being allowed to dry for two weeks in a temperature-humidity controlled room before being sealed with epoxy sealer on the four vertical sides. Afterward, a 75 mm (3 in.) wide and 150 mm (6 in.) long plastic dam with a minimum height of 75 mm (3 in.) was installed on top of the beam. 

Eight 76 × 102 × 406 mm (3″ × 4″ × 16″) specimens were also prepared in accordance with ASTM C666 [36] requirements in order to investigate the concrete’s resistance to freezing/thawing cycles. After being removed from the molds, specimens were cured for 21 days in a water curing tank (23 ± 2 °C in temperature). Ten cylinders of Φ100 × 150 mm (Φ4″ × 8″) were constructed for the control and CWA modified concrete in the self-sealing experiment. All cylinders were kept in a laboratory and cured at ambient temperature prior to cracking.

### 2.3. Items of Investigation

#### 2.3.1. Steel Reinforcement Corrosion

The testing process for all samples at the same age began once the beams were totally cured. Beams were exposed to wet/dry cycles with NaCl solution (approximate volume: 400 mL) for every two weeks at ambient room temperature. The following methods were used for testing the specimens.

#### 2.3.2. Corrosion Potential (Half-Cell Potential)

Half-cell Corrosion Potential (HCP) measurements, following ASTM C876-15 [34] methodology, were taken by commercially available non-destructive Gamry Reference 600+ npotentiostat, Gamry Instrument, Warminster, PA, USA. The rebar embedded inside concrete, and the copper-copper sulfate electrode (CSE) acted as working electrode and reference electrode, respectively. The corrosion potential of each sample was measured three times at the same location and the average was used as the final value. HCP is a useful tool for measuring corrosion potential of steel rebar, but it is important to note that this method is not an appropriate testing technique to obtain any information regarding the rate of corrosion. Therefore, it is usually used in tandem with another corrosion monitoring method.

#### 2.3.3. Macro-Cell Corrosion Rate

When the concrete beams are exposed to chloride ions and high moisture content, their resistivity can decrease, making it possible for anodes and cathodes to be separated [47]. Hence, the type of corrosion observed is typically macro-cell corrosion where top reinforcement acts as anode while two bottom rebars act as cathode in this case. The potential difference between the anode and cathode causes current flow and corrosion. This corrosion mechanism was mimicked with properly designed samples containing two layers of reinforcement that were electrically connected through a resistor (R = 100 Ohm). The corrosion current flowing between the reinforcement layers was determined by measuring the voltage drop over the resistor. Testing of each sample was continued until 48 weeks even though standard mentions that testing can be terminated when the average integrated current over time is 150 coulombs or greater. The ASTM G109 standard [35] also notes that the time to failure is typically six months, which has been investigated here as well.

#### 2.3.4. Linear Polarization Resistance (LPR)

At different times of exposure, Corrosion Rate (CR), was determined using the electrochemical method, Linear Polarization Resistance (LPR), described in ASTM G 59 [48] and G 102 [49]. LPR is a reliable, nondestructive technique for corrosion measurements [50]. In this technique, the reinforcing steel is polarized by a small amount of potential from its equilibrium potential either potentiostatically or galvanostatically [51]. In this study, the steel reinforcing was polarized potentiostatically by changing its potential with a fixed amount, ΔE, and recording the corresponding current, ΔI, for a fixed duration of 240 s. Since the change in potential, ΔE, is recommended to be in the range of 10–30 mV for obtaining a linear polarization curve [52], an initial potential of 15 mV below the corrosion potential, *E_corr_*, is ramped to a final potential of 15 mV above *E_corr_* at a rate of 0.125 mV/s. The LPR setup that was used for measuring corrosion current is shown in Figure 3. The linear polarization resistance, Rp in Ohm, was determined from the slope of the linear curve obtained by plotting the applied potential (E) vs. current density (*i*) at *i* = 0 (Figure 3b) [48].
(1)$$R_{p}={\left(\frac{\partial \Delta E}{\partial i}\right)}_{i=0,\;dE/dt\to 0}$$
where $\Delta E=E-E_{corr}$. The corrosion current, $I_{corr}$ (in A), was then calculated from the Stern–Geary [52] formula and ASTM G59 [48]:(2)$$I_{corr}=\frac{B}{R_{p}}$$
where *Rp* is polarization resistance (Ohm) and *B* is the Stern–Geary constant (V), given as:(3)$$B=\frac{\beta_{a}\beta_{c}}{2.303\left(\beta_{a}+\beta_{c}\right)}$$
in which $\beta_{a}$ is the anodic Tafel constant and $\beta_{c}$ is the cathodic Tafel constant. For steel in mortar or concrete, a value of B equal to 52 mV (typically $E_{corr}>-0.2\;V$) for steel in passive condition and a value equal to 26 mV (typically $E_{corr}<-0.3\;V$) for steel in active condition are normally used. In this study, B will be equal to 26 mV (if $R_{p}$ and $I_{corr}$ are expressed respectively in kiloohms and microamperes), according to RILEM recommendations [53]. Corrosion current density, $i_{corr}$, was then obtained as follows:(4)$$i_{corr}=\frac{I_{corr}}{A_{s}}$$
where $i_{corr}$ is in A/cm^2^ and $A_{s}$ is the area of exposed steel in cm^2^. Assuming uniform corrosion on the entire rebar surface, the CR in terms of corrosion penetration (μm/year) can be calculated using Faraday’s law as:(5)$$CR=\frac{Ka_{w}}{nF\delta }i_{corr}=\alpha i_{corr}$$
where K=315,360 is a unit conversion factor, F is the Faraday constant ($F=96,485\;C\;{\mathrm{mol}}^{-1}$), n is the number of moles of electrons transferred, $a_{w}$ is the atomic weight in grams, δ is the density of the metal in $g\;{\mathrm{cm}}^{-3}$, and $i_{corr}$ is the corrosion current density in μA·cm^−2^ [54]. A current of 1mA results in a loss of 9.2 g of metal in one year. This can be converted into a volume of metal using the density of steel (7652 kg/m^3^), which in turn can be converted into a loss of section or reduction in diameter if assumptions about the area that this loss is occurring over are made. A current density of 10 mA/m^2^ (1 μA/cm^2^) is equivalent to a metal loss of 11.6 μm/year, where the area being considered is the surface area of the reinforcing bar in the measurement region [55]. Hence, the value of the constant α for steel is considered approximately α_Fe_ = 11.6 μA^−1^ cm^2^ μm year^−1^ for this study.

#### 2.3.5. Freeze–Thaw Resistance of Concrete

##### Freeze–Thaw Resistance Testing (ASTM C666)

In this study, a Humboldt (HC-3186S.4F) freeze–thaw chamber was used to test eight concrete beams simultaneously, with one being control in accordance with ASTM C666 [36]. As per ASTM C666, procedure A (rapid freezing and thawing in water) was used to measure freeze–thaw resistance of samples [36]. This freeze–thaw chamber precisely controls a wide range of temperature fall from 40 °F to 0 °F (4.4 °C to −17.8 °C) and then back to 40 °F (4.4 °C). Two beams of each batch were subjected to 300 freeze–thaw cycles at age of 21 days. 

Required by ASTM C666, mass and dynamic elasticity modulus of beams were also evaluated once after every interval of 30 cycles (every 3–4 days) [36]. Performing these tests is required as supplementary test method to check the freeze–thawing efficiency of any mixes. Moreover, compressive strength of samples was measured every 30 cycles of freeze–thaw using a Schmidt hammer as a Non-Destructive Test (NDT). 

##### Resonant Frequency Testing (RFT) (ASTM C215)

As a part of ASTM C666 [36] the dynamic modulus of elasticity was calculated using RFT according to ASTM C215 [37], in which a concrete beam was stroked by 2 oz spherical head hammer to generate internal vibration/frequency. The generated frequency was recorded/received by an accelerometer (10 mV/g). Eventually, the RFT software was used to measure the longitudinal and transverse frequency of concrete specimen. The experiment was carried out three times for each beam specimen and an average for the resonance frequency was calculated.

##### Compressive Strength Test (ASTM C39 & ASTM C805)

At 21 days from casting—the minimum age requirement to cure the samples in accordance to ASTM C666 [36], three cylindrical specimens (Φ100 × 200 mm) from each mixture were tested in saturated surface dry (SSD) condition, following the procedure reported in ASTM C39 [44] in order to determine each mixture’s base compressive strength before exposure. After every 30 cycles (about 3–4 days), a non-destructive compressive strength test was performed using a Schmidt hammer to assess the strength of beams exposed to freeze/thawing cycles, taking eleven measurements on each face of the concrete beams in accordance with ASTM C805 [56] (Figure 4). Because the Proceq Schmidt hammer was utilized in a vertical downward position, the readings were rectified according to the chart in the Proceq Schmidt hammer manual. For each concrete beam, the mean rebound hammer value is measured and converted to compressive strength.

#### 2.3.6. Self-Sealing Capability of Concrete

After air curing, each sample was placed in a Standard Crack-Inducing Jig (SCIJ) [57] to induce a crack width ranging from 0.1 to 0.6 mm using Universal Testing Machine, MTS Systems, Eden Prairie, MN, USA. Surface crack width for each cylinder (top and bottom sides) was measured by optical crack-detection microscope of AmScope, United Scope LLC., Irvine, CA, USA, at equidistant points along the crack (Figure 5a). The images from crack surface were further analyzed with ImageJ software, ver. 1.531, 2021, National Institutes of Health, Rockville Pike, Bethesda, MD, USA, to determine crack profile/size/width. The measurements were then averaged and recorded as the cylinder average. The cracked specimens were later inserted into single-use molds, special rubber sleeves, sealed using silicon sealant (Figure 5b); then one end of cylinder sample, mold-finished side of cylinder (called bottom), was exposed to a constant water head (~1.7 m). The flow of water through specimens was collected from the top surface of cylinder during casting and measured over a period manually. The water flow through a cracked sample was measured until no leakage or water dripping from it was not observed. Details of the self-sealing test setup are illustrated in Figure 5. 

## 3. Results and Discussion

### 3.1. Steel Reinforcement Corrosion

#### 3.1.1. Fresh and Hardened Properties

Table 2 summarizes concrete compressive strength and fresh properties such as slump and air content. The presence of CWA in the mixture resulted in a slight reduction of a concrete slump while no significant change was observed in the air content. CWA modified concrete also indicated slightly higher compressive strength by 7%, which confirms previous results reported by Azarsa et al. [13]. Overall, the addition of CWA did not show any negative effects on concrete’s fresh and hardened properties.

#### 3.1.2. Visual Observation

After removing the salt solution from the container at the end of the last wet/dry cycles, beams were visually inspected to check for any sign of rust or delamination resulting from steel corrosion. The control beam showed signs of rust within approximately eight months of exposure to the salt solution reservoir area, while no signs of rust were observed in CWA-treated beams until eight months (Figure 6). The corrosion potential and rate for all beams were later analyzed to confirm such observation, indicating the possibility to visually steel corrosion in the control samples.

#### 3.1.3. Half-Cell Potential (HCP) Results

Potential reading is related to the thermodynamics of the corrosion process and determines the probability of corrosion occurring in a particular environment, but cannot evaluate the kinetics of the reaction [59]. Potential measurements for each top rebar (anode) subjected to sodium chloride were conducted in compliance with ASTM C876 [34] using a copper–copper sulfate electrode. The potential variation of the rebar was monitored continuously every 4 weeks over a period of 48 weeks (Figure 7). For control and CWA mixes, the figure represents both individual and average corrosion potential. CWA mix indicates a less than 10% probability of no corrosion for the first 24 weeks. Control concrete, on the other hand, had a greater than 90% probability of not corroding within the first 8 weeks. The addition of CWA to concrete slowed the corrosion of reinforcing. According to Table 3 taken from ASTM C876 [34], a potential drop to −350 mV_CSE_ or lower indicates that there is a greater than 90% probability of corrosion risk for reinforcing steel embedded in concrete. While active corrosion took place for the control beams after 28 weeks of exposure, the corrosion initiation for CWA-treated samples happened after 48 weeks of exposure. This could be due to either a reduction in solution permeability into the concrete matrix (due to the presence of CWA crystals) or a progressive restriction in the supply of oxygen to the surface of the embedded reinforcing steel (due to void reduction and interconnectivity of pores for oxygen molecules transfer), or a combination of the two [60].

#### 3.1.4. Macrocell Corrosion Results

As per ASTM G109 [35], the total charge passed through the resistor between the two layers of reinforcement was monitored for both sample types, and samples were identified as being active when a total charge of 150 Coulombs or more passed through the resistor, as illustrated in Figure 8. The total charge passed for each sample in 48 weeks is presented on the left side of the graph, while the average value for each mix is shown on the right side. The macrocell current and hence the total charge travelling through the samples at the start of the test are very low, as seen in Figure 8. When the total charge going through the resistor exceeds 150 Coulombs, both begin to increase at the point of activation. The average active corrosion status (above 150 Coulombs) for control samples occurred at 32 weeks of exposure, whereas this active stage was on the 44th week for beams with CWA additive, indicating admixture effect on delaying corrosion onset. After 48 weeks of exposure to sodium chloride solution, the testing was completed. The data analysis revealed that the control mix had a higher rate of macro-cell corrosion than the CWA mix after the testing was completed.

#### 3.1.5. Corrosion Current Density and Rate Results

The measured values of corrosion current densities (*i_corr_*) and corrosion rate (CR) using LPR method at different time intervals of the exposure period are illustrated in Figure 9. Based on the criteria for the state of corrosion of steel in concrete, as shown in Figure 9, it is observed that control mixes were in the state of moderate to high corrosion on the 32nd week, while this occurred to CWA mixes on the 40th week. As the *i_corr_* for control samples far exceeded the passive limit of 0.1 µA/cm^2^ at an earlier stage, this indicates the lower corrosion rate for concrete beams treated with CWA due to less available space and impermeable environment for ions’ movement. A variety of factors influence the measured *i_corr_* values, such as localized pitting corrosion, testing and physical conditions, which cause fluctuations in the CR values over time [51]. Considering the results of untreated samples, it can be confirmed that the addition of CWA to the mixtures certainly reduces the rate of corrosion, as it provides lower porosity and pore interconnection, resulting in blocking ions pathway from anodic to cathodic site. Considering both control and CWA mixes, it was not possible to pick up the exact time of corrosion initiation from LPR technique (this is the testing technique limitation); hence, it is difficult to point out which one of the two mixes would have longer initiation period under the simulated harsh test conditions. Although corrosion initiation time cannot be precisely determined with LPR, but the results indicate that the presence of CWA in the mix delays the initiation. LPR technique is greatly useful to estimate the rate of corrosion. While Figure 9—left side shows each sample’s corrosion rate and current density over the course of 48 weeks, Figure 9—right side illustrates that the slope of the line passing through average values of corrosion current density and CR, is greater for control samples, representing slower corrosion rate in CWA-treated specimens.

Gamry Reference 600+ measures the HCP, and presents it as corrosion potential, *E_corr_*. Generally, the more negative the *E_corr_*, the higher the *i_corr_*, although large scattering of data occurs (Figure 10). Categorized by Broomfield [61] as boundary value for the low-to-moderate corrosion rate measured by using confined current technique, *i_corr_* was approximately 0.1 µA/cm^2^ at an *E_corr_* of about −300 mV. The reinforcement corrosion activity in the current study largely supported this hypothesis (Figure 10). For both untreated and CWA-treated beams, Figure 10 illustrates an approximately linear relationship between the logarithm of corrosion current density, macrocell corrosion, and the corrosion potential. Considering all corrosion testing results being used in this study, divided in three-time steps (0–16, 16–32, and 32–48 weeks), the higher corrosion probability and rate (about 30% higher) was observed for control specimens within 32–48 weeks of exposure when compared with CWA-modified mixes. This has been circled in Figure 10 where total charge passed is above 150 C, the corrosion potential is lower than −350 mV and *i_corr_* is greater than 1 µA/cm^2^. As can be seen, most of points within high corrosion risk belong to control samples and it can be stated that CWA mixes did not show signs of severe corrosion within the testing period (48 weeks), indicating its influence on delaying/avoiding corrosion in reinforcement.

#### 3.1.6. Freeze–Thaw Resistance of Concrete

##### Fresh and Hardened Properties

Table 4 summarizes fresh properties such as slump, and in addition, compressive strength of concrete samples treated with PRA. As expected, addition of air-entraining agent increased the air content and slightly reduced compressive strength of concrete samples. Also, the density of concrete mixtures with AEA decreased by less than 5–7%. Generally, the addition of the admixture (CWA) into concretes with and without AEA did not substantially alter fresh properties and compressive strength of these mixes.

##### Visual Observations

Figure 11 illustrates the surface of concrete beams after being exposed to 300 cycles of freeze and thawing (at approximately 30 days after placing specimens in the chamber). The surface layer of control-NAEA beams were roughened through these cycles with signs of scaling as can be noticed from left image (Figure 11a) in the figure while the same set with air entraining admixtures does not indicate any significant sign of deterioration. In addition, visual comparison between AEA and non-AEA samples for CWA admixtures does not show scaling or roughening sign on their surfaces. It should be also noted that one of the CWA samples got damaged at 240 cycles during handling for testing and its subsequent results since then have been removed from this report. 

##### Compressive Strength Results Using Schmidt Hammer

As expected, the compressive strength of all mixtures decreased over 300 cycles of freeze–thaw (approximately 30 days) due to exposure to harsh freeze/thawing environment (Figure 12). It should be noted that there are several factors that may result in a slight increase/decrease in the compressive strength of concrete including moisture content, accuracy of testing technique, surface preparation, etc. So, it would be more reliable to investigate the effect of other factors on compressive strength of concrete containing the admixtures. It should also be noted that the compressive strength results obtained from Schmidt Hammer Test (SHT) were noticeably different from actual compressive strength test. As reported by Tarranza and Sanchez [61], the SHT is not an ideal substitute for developing the actual compressive strength of concrete and is influenced by a number of factors that affect surface hardness, e.g., moisture content, age, surface smoothness, carbonation, and temperature of concrete. 

However, SHT has a couple of advantages including those of an NDT method, easy to use, indication of uniformity properties of the surface, inexpensive apparatus/method.

##### Mass Loss

Figure 13 shows the average mass loss of both NAEA-based and AEA-based concrete samples increased over 300 cycles of freeze–thaw. The rate of mass loss of control concrete samples was higher compared to treated samples. Also, addition of AEA into mixtures resulted in substantial improvement in mass reduction of these samples, showing mass loss percentage lower than 1% for AEA samples while NAEA samples are placed 1–4%. Observing from Figure 13, it can be concluded that the rate of mass loss for non-air entrained CWA samples is almost similar to those control samples with AEA indicating addition of CWA can be beneficial in this regard. NAEA-based control samples had considerably higher rate of mass loss (~4%) compared to NAEA-based concrete mixed CWA (~1%) admixtures (Figure 13)) while the range of mass loss for all AEA-based concrete samples is 0.5–1% (Figure 13b).

##### Relative Dynamic Modulus of Elasticity (RDME) of Concrete

Figure 14 indicates the average RDME values measured on two beams versus number of freeze–thaw cycles based on ASTM C215 [37]. According to ASTM C666 [36], unacceptable RDMEs are defined as a RDME below 60%. That is why the specified number of cycles were selected at which the exposure to freeze–thaw was terminated. In the current study, RDME values for control NAEA based mix falls below 60%, indicating rejection of this mix application for freeze/thawing environment. Measurements have been taken from two control-NAEA beams, with RDME equal to 57.1% and 72.9% (one is below 60%). The remaining mixes are still in acceptable range, although their RDME values are different from each other. Admixtures treated samples had less rate of RDME reduction than control ones (about 29% and 9% lower for non- and air-entrained mixes). So, it can be concluded that recommended dosage of admixtures in concrete mixture can increase the freeze–thaw resistance of concrete. In general, RDME values of all types of samples decreased with increasing number of freeze–thaw cycles. This reduction in RDME values mostly attributed to the increased amount of water in capillary pores that cause crack initiation in microstructure of concrete samples [62,63]. According to Yu et al.’s study [63], concrete’s freeze and thaw service life ratio of laboratory testing and natural environment exposure was 1:8–1:9 for every cycle. This means that 300 cycles in the laboratory accelerated testing condition translates to 2400–2700 cycles in natural environment. Considering every cycle takes a day in freeze–thaw natural environment cycles, the concrete can roughly withstand in desired service life for 2700 days or 30 years (3 months in each year experiencing FT cycles against FT environment. To put it another way, every 300 cycles (or 30 days) in a laboratory testing chamber can represent approximately 30 years of concrete service life in a natural freeze–thaw climate. This is good indication of performance of these admixtures in real natural environment as accelerated testing conditions are typically being considered in the laboratory and no failure sign was observed for treated specimens. As mentioned earlier, one of the CWA-AEA samples was excluded from this study due to damage during transportation.

##### Durability Factor (DF)

Basically, the Durability Factor (DF) of a material needs to be calculated when the properties of material are changed (with time), where there is potential of deterioration because of exposure. This has been defined as the average of RDME values multiplied by the number of cycles at which the sample failed (RDME dropped to 60% of its initial value), then divided by the total number of cycles (300 cycles in this case). DF of concrete samples, calculated from the procedure mentioned in ASTM C666 [36], are summarized in Table 5. In the second column, it shows the defined range of DF in ASTM corresponding to a certain standard deviation and acceptable range of the results. The average DF of NAEA-based concrete samples ranged from about 62% to 92%. While the average DF of AEA-based concrete samples ranged from about 87% to 95%. Also, according to the study by Zaharieva [64], the freeze–thaw resistance of concrete samples can be divided into the following classifications: freeze–thaw nonresistant (0% < DF ≤ 40%), unverified freeze–thaw resistance (40% < DF ≤ 60%), traversable freeze–thaw resistance (60% < DF ≤ 80%), and freeze–thaw resistant (80% < DF ≤ 100%). The status of DF for each mixture is shown in column 3 and the values in column 4 are the corresponding RDME failure status according to RDME results (Table 5). It is evident from the results of DF that with one exception (mix NAEA-based control sample), admixture-based concrete samples including control AEA-based samples obtained a higher DF which can only be attributed to the existence of admixtures in concrete.

Finally, it is worth noting that a concrete structure subjected to freeze–thaw cycles is more susceptible to corrosion, and vice versa. Although the combined effect was not evaluated in this study, CWA demonstrated positive performance in an individual deterioration mechanism that can conclude an improvement in a structure’s overall performance even when exposed to both environments at the same time.

#### 3.1.7. Self-Sealing Capability of Concrete

##### Microscopical Observations

ImageJ software and related studies were used to measure the crack width and profile on the surface of cracks using an optical microscope [65,66,67,68]. By stacking all images from a surface, Figure 15 illustrates the crack profile, taken from the top and bottom of concrete cylinders, for both CWA and control samples before exposure to testing conditions. It should be noted that the bottom of a cylinder in a mold was subjected to water pressure from the tank. These images were isolated and converted to binary images using ImageJ, determining their crack width as shown in Figure 16b. The first observed aspect of the crack sealing phenomenon was white crack-sealing formation in control and CWA-treated cylinders, which can be clearly seen in Figure 16 and Figure 17. It can be further observed that CWA specimens have a greater covered area, closing cracks almost completely. Crack closing was also observed on the top surface (the end of sample where water was flowing of) in control specimens, though to a lower extent. These findings confirm the role of CWA admixture in the crack-closing, which leads to the sealing process to a higher degree. A qualitative evaluation of the white precipitant forming on the crack was performed for CWA specimens (Figure 17). The purpose of this evaluation was to discern whether those products are mainly pore-blocking crystals. The close view of crack indicated that the crystal products were formed on the surface of crack as shown in Figure 17c.

#### 3.1.8. Crack Width and Flow Measurement Results:

The surface crack width (mm), measured initial flow, percent flow-reduction rate, and Sealing Ratio (SR) $\left(1-\frac{flow\;at\;time\;\left(t\right)}{initial\;flow}\right)$, were quantified and determined as summarized in Table 6. The average crack width of three cylinders is about 0.36 mm, being very similar between two mixtures. Water flow is also considerably reduced in control samples (86–92 percent), but it is entirely stopped and reached a 100 percent flow reduction in CWA-treated samples after around 25 days of testing. Similar results were obtained for SR values, indicating better performance of CWA mix in this regard. It is concluded that even though the surface crack width was very similar for both mixes, but it did not truly represent the sealing capability between the mixes since the higher flow reduction, complete water stop, and higher sealing ratio were obtained for CWA samples. This can be mainly due to formation of additional pore-blocking crystals through crack, resulting in blocking water pathway and closing crack path. 

The results of flow rate and self-sealing ratio for all samples and their average values are illustrated in Figure 18. The initial water flow through samples was 6.500–10.050 mL/h for CWA-treated samples while this value was within 4.800 to 8.400 mL/h for control cylinders. Although the average crack width for both specimens is very similar but the initial flow rate is higher for CWA specimens. This confirms that equivalent crack sizes of CWA samples, proposed in [32], are greater than control mixes while their actual crack width is the same. In other words, two samples with the same surface crack width do not necessarily have the same water flow rate. One can observe that the cracked cylinders experienced a rapid initial flow during the first 100 h and reached to flow rate below 3000 mL/h, whereas the water flow reached to different reduction rate and a steady state after 300 h. Overall, the water flow through the samples reduced over time, indicating “self-sealing” of concrete. 

The addition of CWA into mix yielded positive results with regards to self-sealing as the flow rate dropped to zero for two cylinders within 20 days, indicating complete sealing. Within the monitoring period, the control samples did not show complete water stop age and had an above zero flow rate at all times. As discussed in the previous section, the main reason for absolute water-stop age and sealing in the CWA samples is most likely due to the deposition of additional crystals into the crack—shown in Figure 18—which acts as a physical barrier, contributing to further filling of the pores and creating sufficient time for concrete to enhance its autogenous sealing process. The self-sealing ratio was also determined for both control and CWA mixes. As shown in Figure 18, the self-sealing enhancement follows quite a similar rate for both samples at very early testing age, but the SS capability and ratio increases at a higher rate for CWA samples after 150 h of testing. This finding is a good indicator of CWA’s effectiveness in sealing a crack in the concrete.

## 4. Conclusions

This paper aimed to draw some outstanding insights about the durability and self-sealing improvement of concretes modified by adding hydrophilic Crystalline Waterproofing Admixtures (CWA), categorized in ACI 212 TC report [4] as Permeability Reducing Admixtures (PRA) for hydrostatic pressure conditions. Based on the findings of this study, the following conclusions can be made:

When exposed to freeze–thaw cycles, greater mass loss and lower durability factors were obtained for mixes without CWA and Air-Entraining Admixtures (AEA). Using resonance frequency testing for performance evaluation, the presence of CWA even in non-air-entrained concrete showed higher dynamic modulus of elasticity value and significant resistance to freeze–thawing cycles.

After 25 days of testing, the surface of cracks in CWA concrete mix has appeared to be filled with sealing materials, both visually and microscopically. Measuring water flow rate through crack indicated that CWA concrete has a greater self-sealing ratio and a faster seal of cracks.

Monitoring corrosion of steel reinforcement for about one year using Linear Polarization Resistance (LPR) technique showed a lower corrosion rate for concrete beams treated with CWA due to less available void space and impermeable environment for ions’ movement.

Active corrosion occurred for the control beams after 28 weeks of exposure, while the corrosion initiation for CWA-treated samples occurred after 48 weeks.

Overall, the use of CWA in a concrete mix improved its crack-closing capability and durability in both corrosive and freeze/thawing cycles exposed environments when compared with a control mix, indicating CWA value as a chemical admixture to be considered in the concrete construction.

## Figures and Tables

**Figure 1 materials-14-06508-f001:**
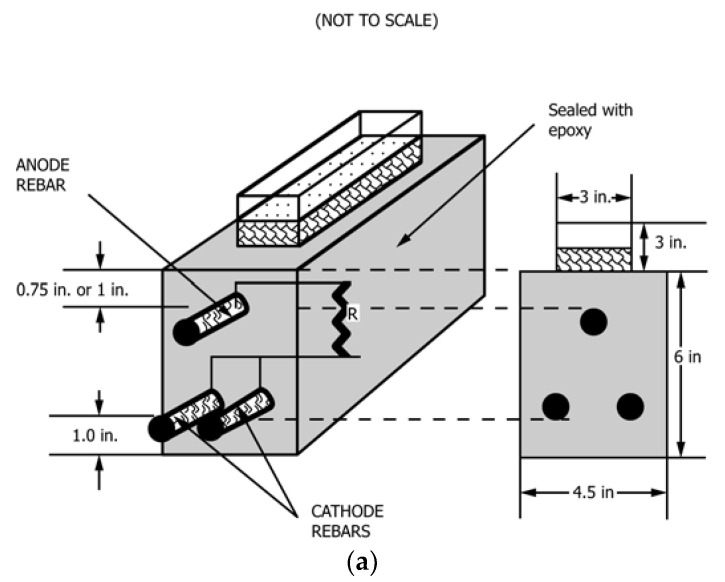
Concrete Beam (All measurements in inches): (**a**) schematic specimen configuration and cross-section view; (**b**) side view.

**Figure 2 materials-14-06508-f002:**
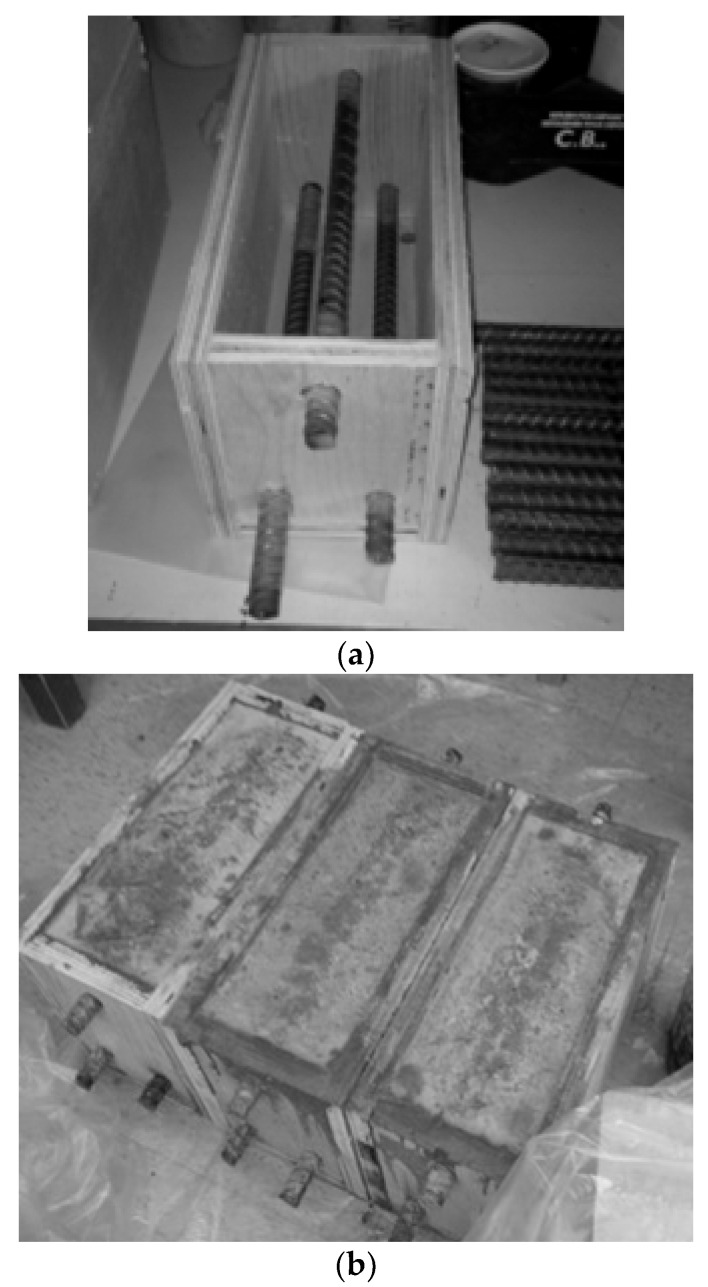
Concrete Beams: (**a**) Specimen mold; (**b**) Prepared specimens for each mix.

**Figure 3 materials-14-06508-f003:**
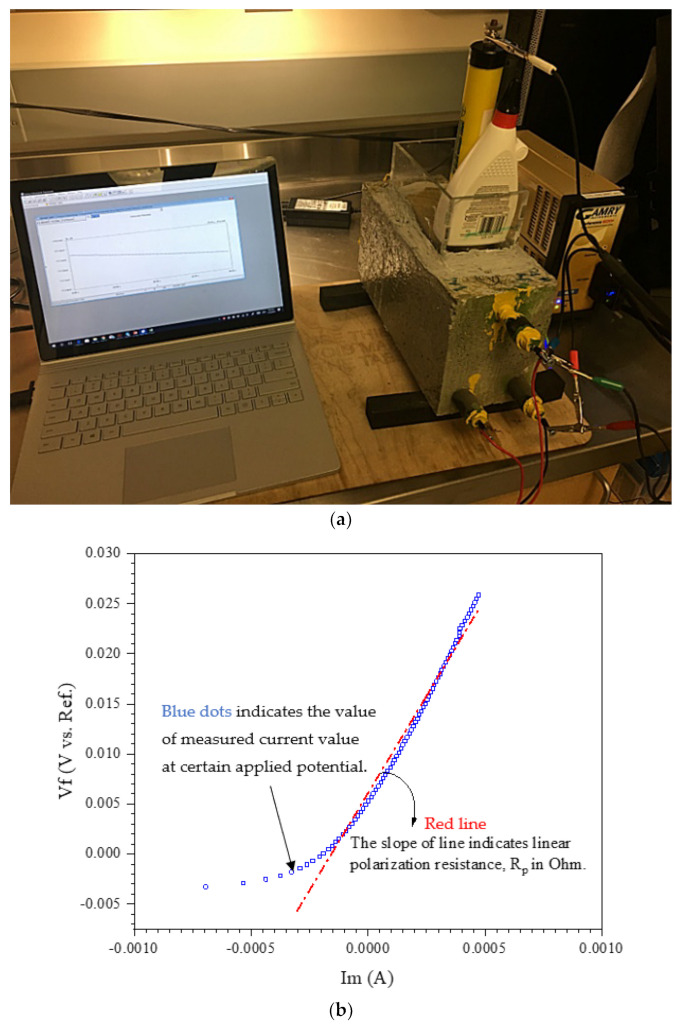
(**a**) Corrosion measurement setup (**b**) applied potential (E) vs. current density (i).

**Figure 4 materials-14-06508-f004:**
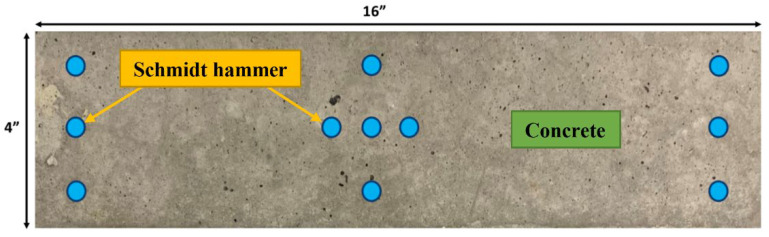
Schematic locations for using a Schmidt hammer.

**Figure 5 materials-14-06508-f005:**
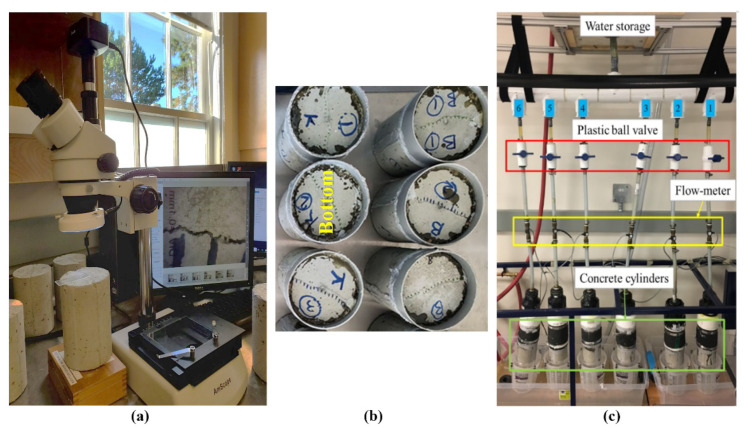
(**a**) Crack investigation using optical microscope. (**b**) Cylinders prepared for testing (bottom side exposed to water flowing from tank). (**c**) Self-sealing setup [58].

**Figure 6 materials-14-06508-f006:**
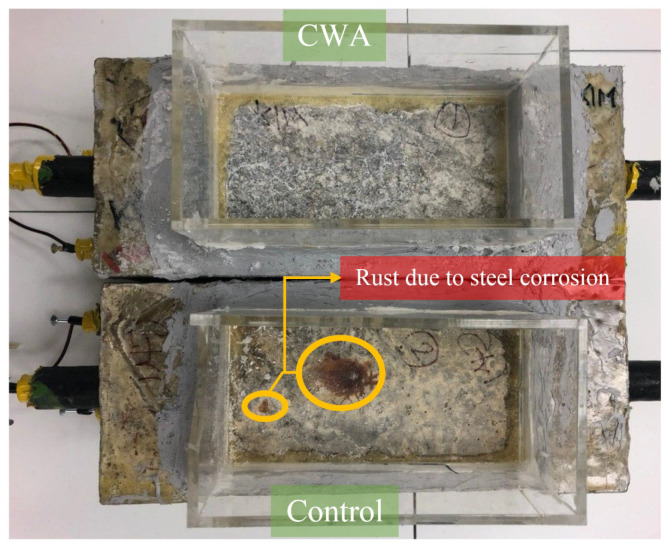
Visual inspection of corrosion in control and CWA-modified concrete beams.

**Figure 7 materials-14-06508-f007:**
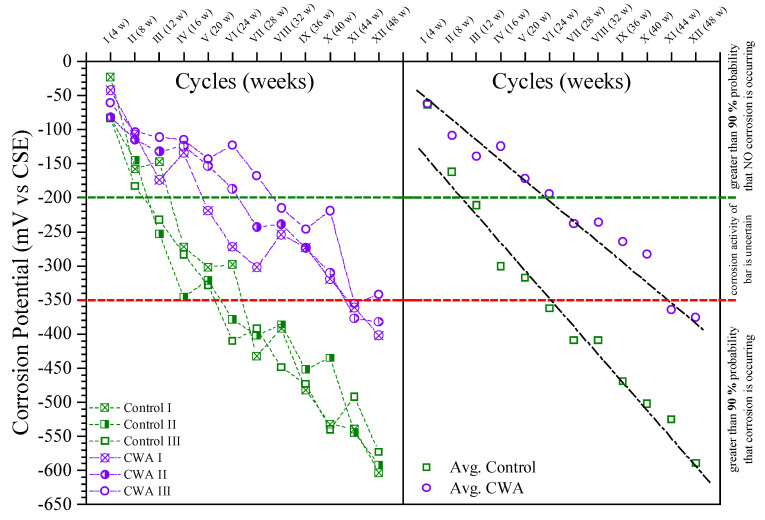
Half-cell potential measurement results.

**Figure 8 materials-14-06508-f008:**
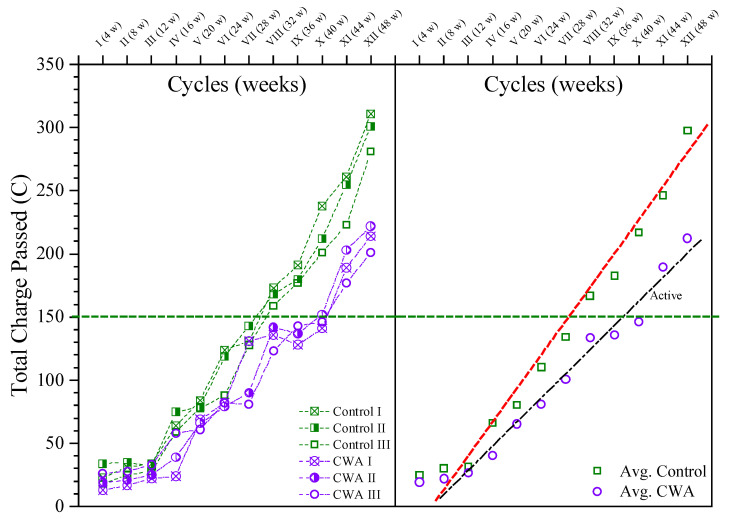
Macro-cell corrosion results.

**Figure 9 materials-14-06508-f009:**
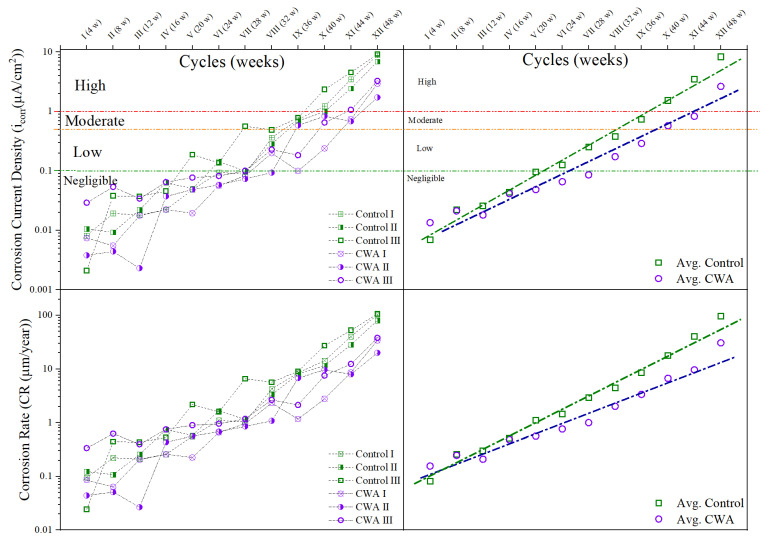
Corrosion current density and rate for reinforced concrete beams exposed to NaCl for 48 weeks.

**Figure 10 materials-14-06508-f010:**
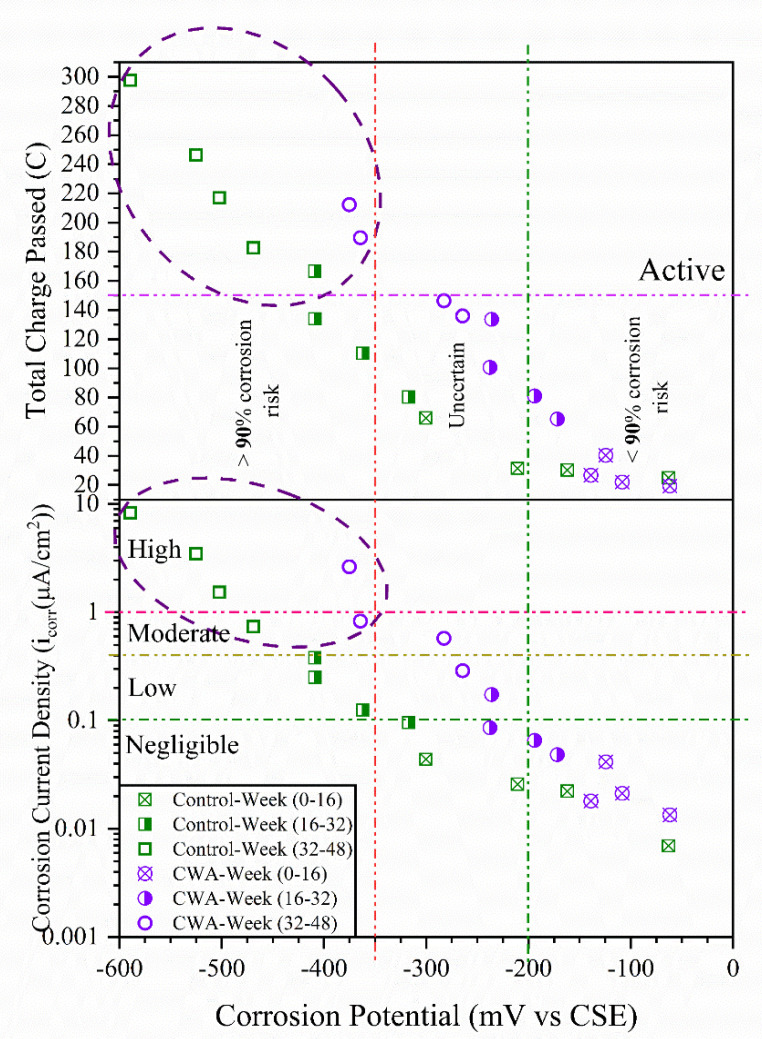
Relationship between corrosion potential, total charge passed and current density.

**Figure 11 materials-14-06508-f011:**
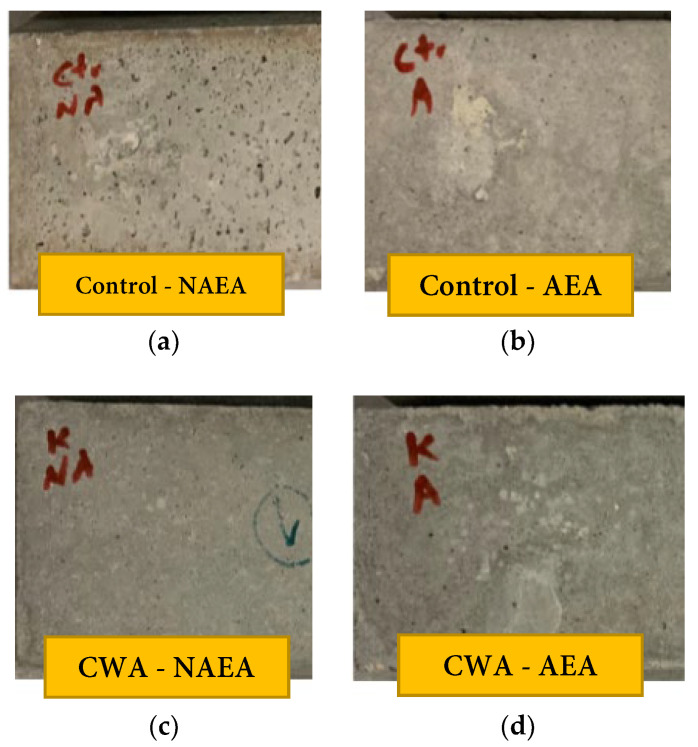
Surface of concrete samples exposed to 300 freeze/thaw cycles: (**a**) Control—NAEA; (**b**) Control—AEA; (**c**) CWA—NAEA; (**d**) CWA—AEA.

**Figure 12 materials-14-06508-f012:**
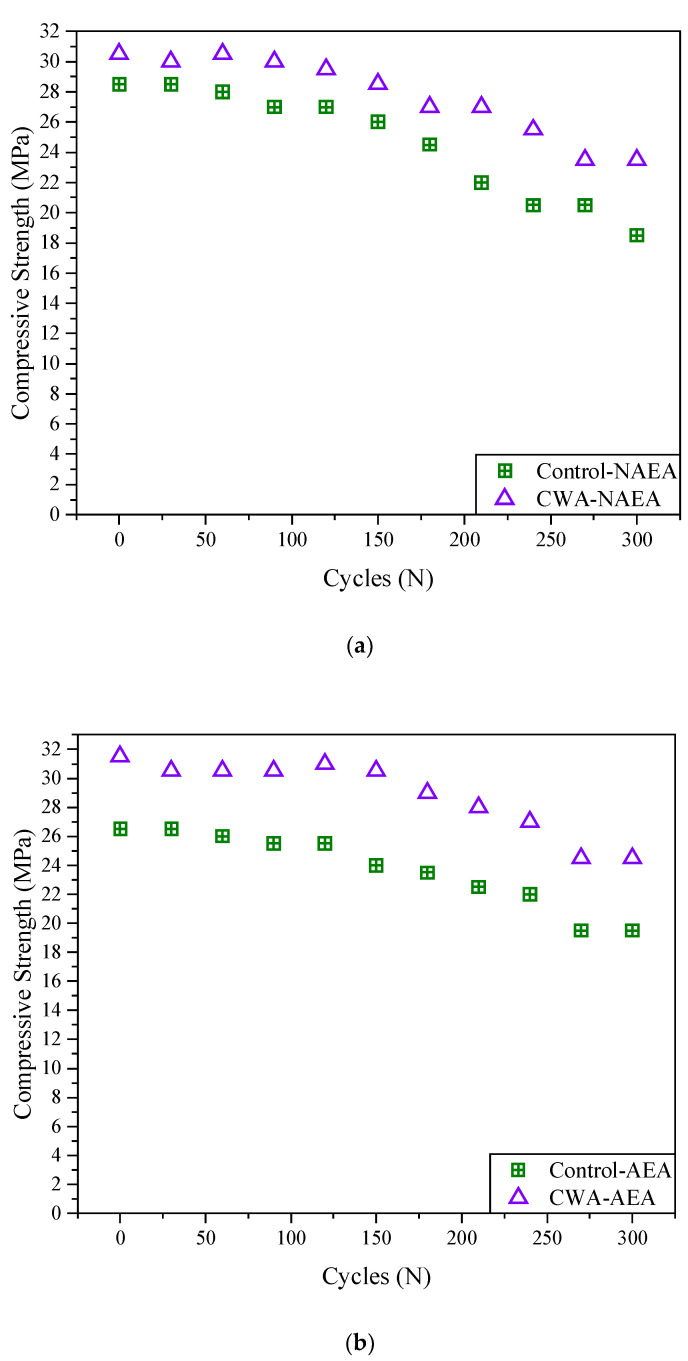
Average compressive strength results obtained using Schmidt hammer test. (**a**) Non-air-entrained mix; (**b**) air-entrained.

**Figure 13 materials-14-06508-f013:**
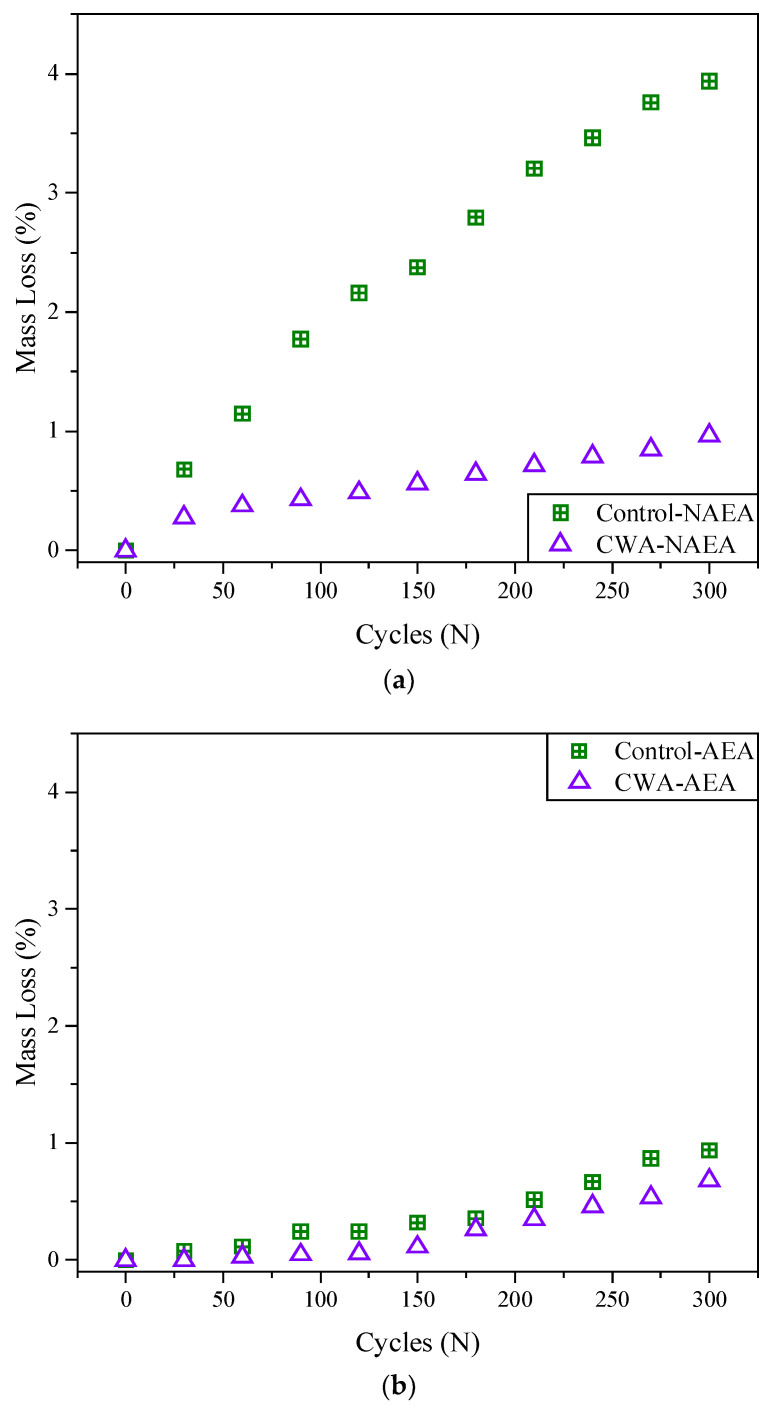
Percentage of mass loss of (**a**) non-air-entrained and (**b**) air-entrained concrete samples.

**Figure 14 materials-14-06508-f014:**
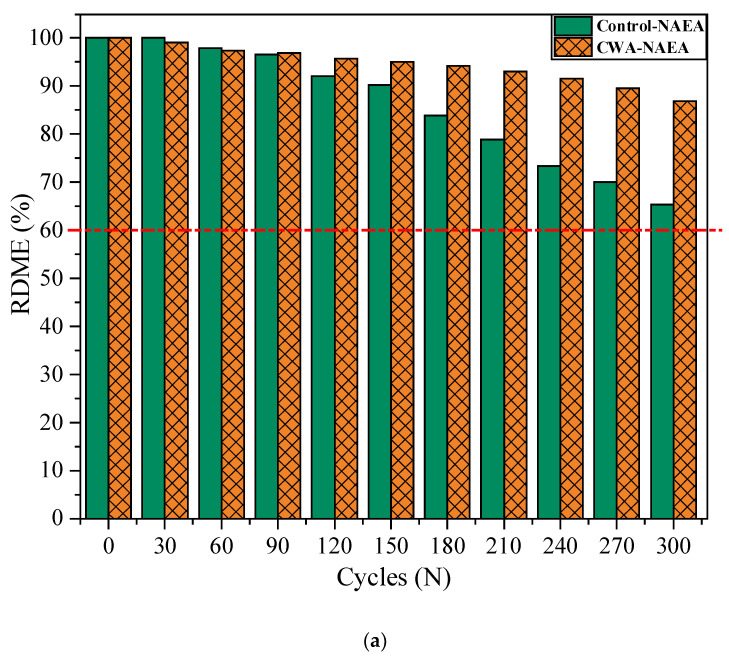
Average Relative Dynamic Modulus of Elasticity (RDME) of concrete beams, (**a**) NAEA CWA vs. Control samples, (**b**) AEA CWA vs. Control samples.

**Figure 15 materials-14-06508-f015:**
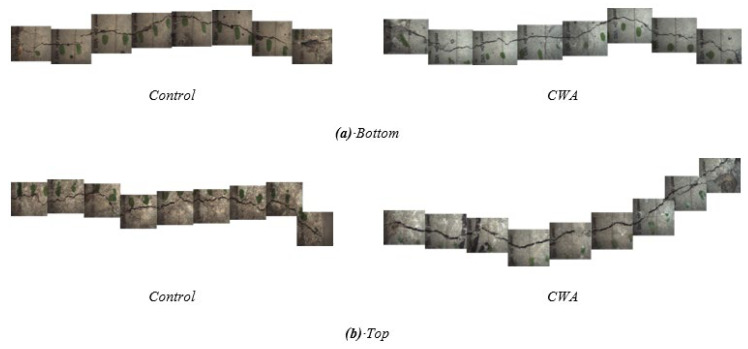
Crack profile of CWA and control concretes (**a**) Cylinder bottom during casting (**b**) Cylinder top during casting.

**Figure 16 materials-14-06508-f016:**
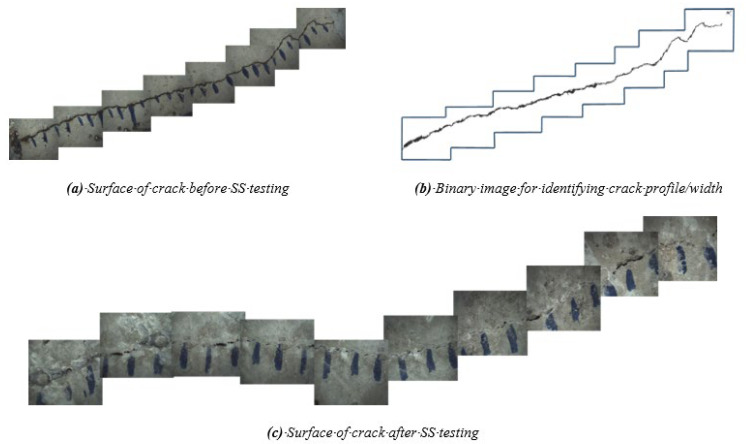
Crack surface of control specimen. (**a**) Before sealing; (**b**) binary image of isolated crack; (**c**) after sealing.

**Figure 17 materials-14-06508-f017:**
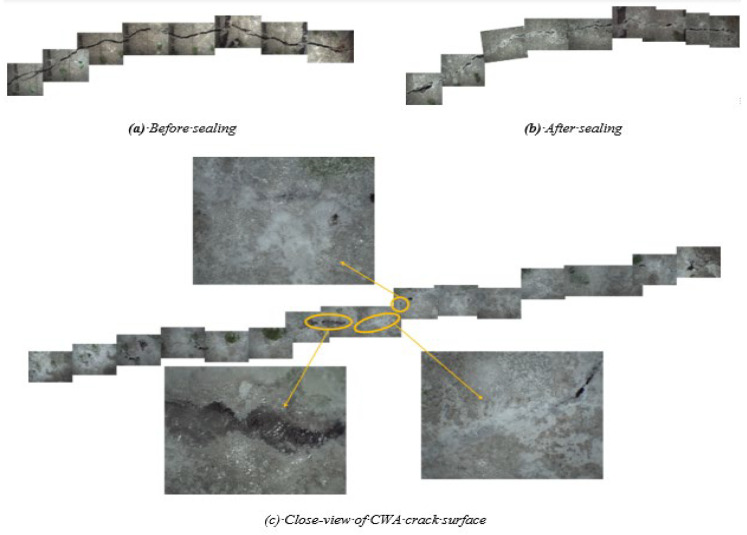
Crack surface of CWA specimen. (**a**) Before sealing; (**b**) after sealing; (**c**) close view of crack surface.

**Figure 18 materials-14-06508-f018:**
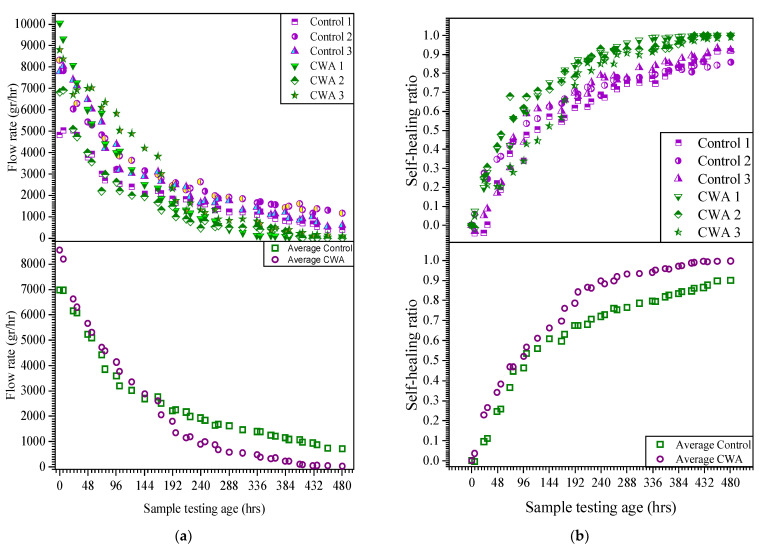
(**a**) Flow rate vs. time; (**b**) self-sealing ratio.

**Table 1 materials-14-06508-t001:** Mix design of control and CWA concretes.

Sample ID/Mix Proportions (kg/m^3^)	Portland Cement	Coarse Aggregates	Sand	Water	CWA
Control	340	1120	820	181	–
CWA					6.8

**Table 2 materials-14-06508-t002:** Fresh and hardened concrete properties.

Sample ID	Slump (mm)	Air Content (%)	Fresh Density (kg/m^3^)	28 Days Compressive Strength (MPa)
Control	125	1.9	2345	31.8
CWA	110	1.8	2328	34.2

**Table 3 materials-14-06508-t003:** Interpretation of corrosion activity of electrochemical methods.

Half-Cell Potential Reading (mV) [34]	Corrosion Activity	Corrosion Current Density (µA/cm^2^) [53]	Mean Corrosion Penetration Rate (µm/Year)	Corrosion Classification	Total Charge Passed (C) [35]	Macrocell Corrosion Status
Copper–Copper Sulfate Electrode (CSE)						
>−200	Greater than 90% probability of no corrosion	≤0.1	≤1.2	Very low or passive	≤150	Passive
−200 to −350	An increasing probability of corrosion	0.1–0.5	1.2–6	Low to moderate	>150	Active
<−350	Greater than 90% probability of corrosion	0.5–1	6–12	Moderate to high	-	-
-	-	>1	>12	High	-	-

**Table 4 materials-14-06508-t004:** Summary of fresh properties and compressive strength of concrete samples.

Sample ID	Air Content (%)ASTM C231 [6]	Density (kg/m^3^)	Average Compressive Strength (MPa)—21 Days (ASTM C39 [3])
Control-AEA	6.8	2241	33.4
Control-NAEA	2.1	2345	36.3
CWA-AEA	7	2079	35.6
CWA-NAEA	1.8	2284	36.8

**Table 5 materials-14-06508-t005:** Durability Factors (DF) for concrete mixes with and without CWA.

Sample ID	Calculated Average DF (%)	Range of Average DF (ASTM C666)	DF Status	RDME Status (Below 60%)
Control-NAEA	62.4	50–70	Traversable freeze–thaw resistance	Fail
CWA-NAEA	86.9	80–90	Freeze–thaw resistant	Pass
Control-AEA	87.1	80–90	Freeze–thaw resistant	Pass
CWA-AEA	93.5	90–95	Freeze–thaw resistant	Pass

**Table 6 materials-14-06508-t006:** Measured crack width and initial flow.

Sample ID		Surface Crack Width (mm)				Real Initial Flow *q*_0_ (mL/h)	$\mathrm{Percent}\;\mathrm{Flow}\text{-}\mathrm{Reduction}\;\mathrm{Rate}\;(1-q_{F}$ $/q_{0})\%$	Sealing Ratio (%)
		Top	Bottom	Average				
Control	I	0.317	0.263	0.290	0.359	4.834	92%	91.9
	II	0.486	0.469	0.477		8.325	86%	85.9
	III	0.328	0.293	0.311		7.793	92%	92.3
CWA	I	0.467	0.275	0.371	0.355	10.045	100%	100
	II	0.400	0.344	0.372		6.822	100%	100
	III	0.331	0.313	0.322		8.799	99%	98.9

## Data Availability

The raw/processed data required to reproduce these findings cannot be shared at this time as the data forms part of an ongoing study.

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
