# Peer review of "Durability and Self-Sealing Examination of Concretes Modified with Crystalline Waterproofing Admixtures"

_materials, 2021, doi:10.3390/ma14216508_

Round 1
Reviewer 1 Report
Please find my comments for the Manuscript "materials-1386940". I suggest that major revisions have to be applied in order to be published in Materials Journal.
The authors are examining the use of a commercial Crystalline Waterproofing Admixture (CWA), and its effect on concrete’s overall freeze-thaw resistance, self-sealing, and corrosion resistance. ASTM standard test procedures are applied for most measurements and the results are reported appropriately. This subject is interesting for its practical applications; however the contribution to the present scientific knowledge is not clear and the methodological approach followed could be part of a technical report. The authors report the results of the standard tests without attempting to explain the scientific reason behind the observations. Chemical composition /mineralogy of the materials used is not reported. Thereby the results cannot be attributed or linked to the mineralogy of the materials used and the study could not be reproduced by peers.
Author Response
The authors appreciate the reviewer's thorough assessment of the manuscript, as well as his or her valuable suggestions. Please find our responses in the attached document.

Reviewer 2 Report
Dear authors,
We are honoured to have the opportunity to prepare a review for this article.
Research in the field of CWA and PRA is significant and necessary.
The structure of the article is good, but a few things need to be improved, especially from a technical point of view, which is given below.
The title is fine, the abstract is a bit longer, but good.
The introduction is stretched without division into chapters.
In connection with the analysis of cracks, I recommend adding, for example:
10.1016 / j.conbuildmat.2018.10.111
10.1016 / j.tafmec.2015.09.005
10.1520 / JTE20170176
Have you reached the required w / c?
On what basis were the two investigated mixtures chosen?
The description of the test methods is OK.
The results are in line with expectations. They can enrich scientific knowledge, but have these results not been found elsewhere?
The conclusions contain concrete results but lack a clear summary of the advantage and significant advantages of the ingredients.
A few notes:
- you need to read the article properly,
- the article contains 39 "error" of references - you need to check it,
- Figure 1 should be divided into two parts - the first part (a and b) should be larger,
- the description and location of the equations do not correspond to the template,
- figure 2 is too small,
- Table 3 is divided,
- Figures 11, 12 and 13 are small and illegible,
Regards,
Author Response

(The authors gave the same response as above.)

Round 2
Reviewer 2 Report
Dear authors,
Thank you for the extensive and detailed responses and corrections.
The article is much better.
Regards,
Author Response
The authors take this opportunity to express their appreciation for your time and valuable consideration in reviewing our manuscript.